# Mutual Coupling Reduction in Compact MIMO Antenna Operating on 28 GHz by Using Novel Decoupling Structure

**DOI:** 10.3390/mi14112065

**Published:** 2023-11-07

**Authors:** Tanvir Islam, Fahd Alsaleem, Fahad N. Alsunaydih, Khaled Alhassoon

**Affiliations:** 1Department of Electrical and Computer Engineering, University of Houston, Houston, TX 77204, USA; tislam7@cougarnet.uh.edu; 2Department of Electrical Engineering, College of Engineering, Qassim University, Unaizah 56452, Saudi Arabia; f.alsaleem@qu.edu.sa (F.A.); f.alsunaydih@qu.edu.sa (F.N.A.)

**Keywords:** mutual coupling, decoupling structure, ECC, MIMO antenna, IoT, 5G mobile communication

## Abstract

This article presents an antenna with compact and simple geometry and a low profile. Roger RT6002, with a 10 mm × 10 mm dimension, is utilized to engineer this work, offering a wideband and high gain. The antenna structure contains a patch of circular-shaped stubs and a circular stub and slot. These insertions are performed to improve the impedance bandwidth of the antenna. The antenna is investigated, and the results are analyzed in the commercially accessible electromagnetic (EM) software tool High Frequency Structure Simulator (HFSS). Afterwards, a two-port multiple–input–multiple–output (MIMO) antenna is engineered by orthogonalizing the second element to the first element. The antenna offers good value for mutual coupling of less than −20 dB. The decoupling structure or parasitic patch is placed between two MIMO elements for more refined mutual coupling of the proposed MIMO antenna. The resultant antenna offers mutual coupling of less than −32 dB. Moreover, other MIMO parameters like envelop correlation coefficient (ECC), mean effective gain (MEG), diversity gain (DG), and channel capacity loss (CCL) are also studied to recommend antennas for future applications. The hardware model is fabricated and tested to validate the results, which resembles software-generated results. Moreover, the comparison of outcomes and other important parameters is performed using published work. The outcome of this proposed work is performed using already published work. The outcomes and comparison make the presented design the best option for future 5G devices.

## 1. Introduction

The swift advancement in communication systems has been recognized since the last decade. The transformation of third generation (3G) to fourth generation (4G or LTE), and then 4G to fifth generation (5G), and now the current deployment of future sixth generation (6G) all demonstrate many changes in communication models and systems [1,2]. These changes are carried out to obtain the required results, which are needed by end-users. The user requires high-speed internet and communication, which are possible with high data rates, throughput, improved link capacity, and the elimination of multipath effects [3]. Another important requirement of any device is a low cost, compact size, and improved battery life. All these requirements lead to advancements and changes in the communication system and model, which further promote changes in devices [4,5].

Being an important and essential component of the communication model, antenna design requirements were revised due to changes in communication devices or models [6]. To meet the aforementioned demands from users, the design challenges for engineering antennas have changed. For both present 4G or 5G and upcoming communication systems (6G), a compact and simplified antenna with a low profile is required to integrate easily with other communication system parts [7,8]. Besides the geometrical requirements, changes in the performance of antennas are also in demand. An antenna with high gain, wideband, and high efficiency is required to achieve a high data rate. And, in the case of MIMO, antenna isolation and other MIMO parameters are important [9].

Researchers and academics have suggested a variety of antennas for this purpose that operate at high gain, wideband, and compact sizes [10]. Multi-input multi-output (MIMO) antenna technology is used to operate at elevated data rates and raise link capacity [11]. Mutual coupling between MIMO antenna elements is a crucial MIMO antenna characteristic [12,13]. The antenna with MIMO configuration is self-isolated, which means that the antenna elements are not affected by each other and offer a transmission coefficient below the required value of <20 dB [14].

In many MIMO antennas, the required value of mutual coupling is not met, for which various techniques are used to obtain the mutual coupling below 20 dB. Researchers have adopted many techniques to improve the isolation between elements in MIMO antenna systems. These techniques include the loading of FSS (Frequency Selective Surface) sheets [15,16], absorbers [17], metamaterial [18,19,20,21,22], defective ground structures (DGS) [23,24,25,26], and decoupling structures [27,28,29,30,31]. The FSS sheet is kept below the antenna to bounce back the waves in order to upgrade the outcomes. Due to the bounced-back radiation and antenna radiation, the overall performance is improved [16]. In the case of an absorber, an unwanted spectrum is eliminated to improve the function of the desired band. The antenna results in terms of the desired frequency range are polished, and are then utilized for communication [17].

In the literature, the metamaterials, the defective ground structure (DGS), and the decoupling structures are widely used to upgrade the outcomes of the MIMO antenna in terms of mutual coupling, bandwidth, and gain. To enhance mutual coupling, the antenna is loaded with metastructures, either of one particular layer (same layer or substrate) or a distinct one (load another substrate) [18]. The antipodal Fermi-tapered slot antenna, which has an entire dimension of 85 mm × 21 mm × 0.508 mm, is discussed in [19]. The antenna’s maximal gain and mutual coupling are 10.5 dBi and 38 dB, respectively, and it works over a spectrum of 27–32 GHz. Although the loading of metamaterial improves antenna isolation, its size is large, and its shape is quite complex. Another small and compact size, metamaterial-loaded antenna is provided in [20]. The antenna is 26 mm × 14.5 mm in overall size and operates in the 26.5–29.5 GHz frequency range. The offered antenna gives 38 dB of mutual coupling at a resonant frequency and has two ports. The antenna has a complex geometric structure and low gain despite its small size and good mutual coupling value. 

In [21,22], a high-isolated antenna is given, which offers isolations around 40 dB. The antenna performs at 25–26.5 GHz bandwidth and contains four elements. The setback of this work is the complex design structure due to the array antenna and metastructure, as well as the narrow band. In [23], the DGS-based array antenna is given for millimeter-wave implementations. The antenna offers 1 GHz bandwidth and an ECC of 0.0003. The antenna operating over millimeter-wave frequency is required to operate over wideband. The antenna reported in [24,25,26] contains a DGS ground plane and has simplified geometry. These works show that DGS improves the isolation of reported works, but these antennas have setbacks of either offering a narrow bandwidth or a high value of ECC.

The most efficient technique employed to lessen mutual coupling between MIMO antenna elements is loading decoupling structures between MIMO elements, or parasitic patches/element [27]. In the literature, a number of antennas are reported, where this technique is adopted to improve the isolation of the antenna. The antenna presented in [28] has a compact size of 20 mm × 20 mm × 0.524 mm and offers an impedance bandwidth of 0.75 GHz ranging from 27.25 to 28.5 GHz. The antenna provides mutual coupling of 24 dB after loading the parasitic patch, and it has complex geometry as well as a narrow bandwidth. Another compact design is reported in [29], which has overall dimensions of 28 mm × 28 mm × 0.79 mm and operates over 26.5–31.5 GHz. The reported design has a mutual coupling of 35 dB after loading the parasitic patch. 

A DRA antenna loaded with a parasitic patch is given in [30], which offers measurements of 25 mm × 15 mm and a bandwidth of 26–29 GHz. The mentioned work has a mutual coupling of 25 dB and an ECC of 0.007. A monopole antenna reported in [31] operates at a wideband of 20–28 GHz with an isolation of 15 dB. The value of isolation offered by this design is not under the standard value of mutual coupling required (which should be less than −20 dB). A dual-layer decoupling structure-loaded MIMO antenna is presented in [32]. The antenna offers an impedance bandwidth of 0.8 GHz and contains simple geometry.

A thorough study of the literature reveals that there is still a need for research to develop antennas with small sizes, straightforward geometries, and low profiles, as well as strong gains, wideband, and low mutual coupling. Due to this fact, a unique and straightforward decoupling structure for the antenna is described in this study in order to eliminate mutual coupling. The presented design of an antenna has the following advantages over other work presented in the literature:
Compact size and simple design structure;Wideband and high gain;Simple and new decoupling structure;Low mutual coupling;Low value of ECC and acceptable value of other MIMO parameters.

The rest of the paper is divided into sections. In next section, the single element of the antenna is discussed along with its results. The fabricated antenna prototype and comparison between measured and simulated results are given in Section 2. In Section 3, the MIMO antenna with and without a parasitic patch is discussed. The antenna performance, working, fabricated prototype, and results are given in this particular section. The comparison of proposed work is also provided with already published work in the literature. At the end, the proposed work is concluded and references are provided. 

## 2. Study on Single Element of Antenna

The geometrical and structural layout of the presented dual-port MIMO antenna is given in Figure 1. The antenna contains a simple structure with two circular patches with a circular slot and coaxial feeding. The antenna is placed on the front side of the substrate Roger RT/Duroid 6002 and has an overall compact size of A_X_ × A_Y_ × t = 10 mm × 10 mm × 1.52 mm. The antenna is excited using coaxial cables with 50 Ω impedance matching. The coaxial wire connects to the antenna after passing through the substrates. Using a coaxial cable has the advantages of low cost, being easy to wire and expand, and being supportive of high-bandwidth signals [33]. The rest of the antenna parameters are as follows: R_1_ = 3 mm; R_2_ = 2.7 mm; R_3_ = 2.5 mm; F_X_ = 3.5 mm; F_Y_ = 0.5 mm; t_1_ = 1.52 mm. The antenna is analyzed, and its parameters are studied using an electromagnetic (EM) software tool, High Frequency Structure Simulator (HFSSv9). 

### 2.1. Antenna Design Steps

In this manuscript, a broadband antenna with an efficient value of return loss is proposed for future 5G devices. Desired and beneficial outcomes are achieved after going through three design steps. The design process of the proposed antenna is divided into the three stages shown below:In the initial design stage, a circular patch antenna with a coaxial feedline is engineered for 28 GHz applications. The radius of the circular patch R_1_ = 2 mm is obtained using the circular patch antenna equation given in [34]. The antenna operates at 27.5–28.2 GHz, with a resonant frequency of around 27.9 GHz. The antenna offers a return loss of 20 dB at resonant frequency.To upgrade the bandwidth and refine the return loss of the antenna, another circular stub is loaded onto the antenna, which has a radius of R_2_ = 2.5 mm. After this step, the antenna starts resonating at 28.2 GHz, with an impedance bandwidth of 27.1–29.2 GHz. The value of return loss also improved after this step and approached 27 dB.For further improvement in results, a circular-shaped slot with a radius of R = 2.5 mm is etched from the antenna. After this step, the final structure of the antenna is obtained, and it starts operating at a broadband of 25.25–29.85 GHz. At this stage, the antenna offers a return loss of –45 dB.

The antenna design evolution along with its impact in terms of the S_11_ plot is given in Figure 2a,b, respectively.

### 2.2. Antenna Single-Element Results

The performance evaluation of an antenna is analyzed by studying its S-parameter curve and gain versus frequency curve. It is noticed from Figure 3a that the antenna operates at 27.85 GHz with an operational bandwidth of 4.6 GHz ranging from 25.25 to 29.85 GHz. The antenna offers quite a high gain of greater than 9 dBi at operational bandwidth. The antenna offers a peak gain of 10.8 dBi at a resonant frequency of 27.85 GHz. To cross-check and validate the simulated outcomes, the hardware model is fabricated. The result of the prototype is also added to the figure, which shows strong agreement with software-generated results. The offered outcomes in terms of S11 and gain show that the antenna is a good applicant for future 5G devices.

The radiating property of the presented broadband design is examined by studying the radiation patterns of proposed antenna. The radiation pattern of a single element of this work is provided in Figure 4, at a resonant frequency of 28 GHz. The illustration shows that the antenna operates in the principle H-plane and has a slightly bent pattern in the principle E-plane. For further validation and support of our results, the hardware outcomes of the radiation pattern have also been added to the figure. It is seen that both outcomes have a strong agreement with each other.

## 3. Two-Port MIMO Configuration of Antenna

Figure 5 illustrates the two-port MIMO antenna that is suggested in this paper. The figure shows that the antenna has two MIMO elements that are positioned orthogonally to one another. The antenna is designed on the same material as a single element with a size of M_X_ × M_Y_ = 10 mm × 26 mm. Both of the elements are fed by a coaxial cable and are shown in Figure 5b. Between the two elements of the antenna, a stair-shaped structure is loaded, called a parasitic patch or decoupling patch or structure. The decoupling structure loaded on the antenna has a simple and novel shape with optimized parameters of P_A_ = 10 mm, P_B_ = 1 mm, P_C _= 0.5 mm, P_D_ = 0.5 mm, and P_E _= 5 mm. To lessen the mutual coupling between MIMO parts, the decoupling element is mounted onto the antenna.

For the further validation of outcomes, an antenna prototype is constructed and tested. A snap of the antenna prototype is given in Figure 5. The antenna is fed coaxially. To test near-field outcomes, the PNA network analyzer has the model number N5224A, and, for far-field measurement, an anechoic chamber. Some imperfections in results are observed, which is due to fabrication tolerance as a cheap chemical etching process is utilized; for better results, an LPFK machine with a tolerance of less than 0.01 mm can be utilized. Poor soldering may also be the reason for discrepancy as it is performed by hand; better results can be achieved using machine soldering.

### 3.1. Design Strategy of MIMO Antenna

The antenna is transformed to a MIMO configuration by placing another antenna element orthogonal to the reference antenna. The two components of the MIMO antenna are separated by an 8 mm gap. The antenna has the same optimal characteristics as the single antenna element covered in Section 2.1 of this article. Mutual coupling is the most important characteristic when analyzing the MIMO antenna. It demonstrates how MIMO components interact with one another. To improve the isolation of the antenna, a decoupling structure is loaded between two elements of the MIMO antenna system. With its lowest value being −35 dB at 26.75 GHz, the straightforward two-port MIMO antenna gives a mutual coupling of a little under −20 dB at operational bandwidth. As shown in Figure 6, the stair-shaped parasitic patch is loaded between the MIMO antenna elements to improve the mutual coupling between them. The antenna offers a mutual coupling of less than −32 dB at functional bandwidth after the addition of the decoupling patch, with the lowest value being −48 dB at 27 GHz, as shown on Figure 7.

### 3.2. Reflection and Transmission Coefficient

In Figure 8, the reflection and transmission coefficients are given along with a hardware prototype. As can be observed, the antenna has a resonance frequency of 27.9 GHz and functions over a large frequency range of 25.25 to 29.85 GHz. Around 45 dB of return loss is seen at this value. Additionally, the transmission coefficient of the antenna is shown in the image below. The data show that the antenna has mutual coupling of at least −48 dB at 27.3 GHz and less than −32 dB at active bandwidth. Furthermore, the tested results are also added to the figure to validate the design. It is proven that antennas have the same measured and simulated outcomes with negligible differences. These results make the proposed antenna the best applicant for future compact devices operating on a 5G communication system.

### 3.3. Radiation Pattern

Radiation pattern is the key parameter used to study the radiating properties of a two-port antenna. The antenna’s radiation pattern is shown in Figure 9 alongside the simulated and tested data. It is clear that the antenna offers a broadside radiation pattern in both planes (E and H). The pattern exhibits a slight amount of distortion that results from patch loading. Furthermore, it has been shown that the findings of the simulation and the tests correspond very well. The current surface distribution is given in Figure 9b,c. It can be seen that when port-1 and port-2 are excited, the current induced due to one antenna interacting with another is significantly blocked by the decoupling structure, resulting in low mutual coupling.

### 3.4. Analysis of MIMO Parameters

For more validation of the recommended MIMO antenna design for millimeter applications, some important MIMO parameters are studied and measured. Envelope correlation coefficient (ECC), mean effective gain (MEG), diversity gain (DG), and channel capacity loss (CCL) are some of these parameters.

ECC shows the performance of independent antennas in a MIMO system. Ideally, the value of ECC should be equal to zero or approximately equal to zero and can be calculated using the formula given below [35].
(1)$$\rho_{eij}=\frac{{\left|\iint_{0}^{4\pi }\left[\vec{R_{i}}\left(\theta ,\varphi \right)\times \vec{R_{j}}\left(\theta ,\varphi \right)\right]d\Omega \right|}^{2}}{\iint_{0}^{4\pi }{\left|\vec{R_{i}}\left(\theta ,\varphi \right)\right|}^{2}d\Omega \iint_{0}^{4\pi }{\left|\vec{R_{j}}\left(\theta ,\varphi \right)\right|}^{2}d\Omega }$$

For this design, the value of ECC is less than 0.0001 through an operational bandwidth of 25.25–29.85 GHz, as given in Figure 10a. Moreover, the simulated and measured values of ECC are similar with negligible differences. The offered value of ECC and the results make the proposed antenna a good applicant for future MIMO antenna applications.

Another important MIMO parameter, which is studied in this paper, is channel capacity loss (CCL). The study of correlation losses in MIMO systems is known as CCL. Less than 0.5 bits/Hz/s should be the CCL value, as estimated by the equations provided below [36].
(2)$$CCL=-\log_{2}det\left|\psi^{R}\right|$$
where *ψ^R^* refers to the below matrix for receiving antenna correlation and its mathematical relation is given below.
(3)$$\psi^{R}=\left[\begin{array}{cccc} \rho 11 & \rho 12 & \rho 13 & \rho 14\\ \rho 21 & \rho 22 & \rho 23 & \rho 24\\ \rho 31 & \rho 32 & \rho 33 & \rho 34\\ \rho 41 & \rho 42 & \rho 43 & \rho 44 \end{array}\right]$$

The CCL of the suggested MIMO antenna is shown in Figure 10b. As can be recognized, the operational bandwidth of the antenna is CCL 0.05 bits/Hz/s. Moreover, a good agreement between the software generated and tested results is observed, which makes the proposed work suitable for future communication models.

Diversity gain (DG) is one of the most important parameters of MIMO antenna systems, which measures the losses experienced in the diversity scheme, and is the most crucial metric in MIMO systems. The permissible diversity increase value can be calculated by Equation (4) and its maximum value is 10 dB; however, in real-world situations, a value of roughly 10 dB is acceptable [36]. The mathematical equation for DG is given as
(4)$$DG=10\sqrt{{1-|\rho_{eij}|}^{2}}$$

According to Figure 10c, the suggested antenna provides a DG of approximately 9.99 dB over the operational bandwidth of 25.25–29.85 GHz. Moreover, the simulated and measured results are compared, showing resemblance. The offered value suggests that the antenna is the best candidate for future 5G devices.

Mean effective gain (MEG) is a key metric of the MIMO system, which shows the received power in the fading area. The acceptable range of MEG is less than –3 dB. The proposed work offers MEG less than –5 dB, which is under the acceptable range.

From the above discussion, it is clear that the proposed MIMO antenna offers an acceptable range of MIMO parameters. A comparison between simulated and measured MIMO parameters is also provided. The value offered and the comparison show that the proposed antenna is a good candidate for future devices, which operate at high data rate.

### 3.5. Comparison with the Literature 

The results in the form of bandwidth, gain, isolation, and ECC of the proposed antenna are compared with published work to prove the antenna’s superiority. In the literature table given below, the antenna given in [28,30,31] has a compact size but either offers a narrow band or low isolation. The high-gain antenna given in [19,21] has a large and complex geometry. The antenna referred to in [18,19,20,21,22,23,24,25,26,27,28,29,30] has good isolation but either has complex and large geometry or a narrow band. This conversation demonstrates that there remains a need for research to develop low-cost, low-profile, compact antennas with reduced geometry. High gain, wideband, low value for isolation, and ECC should all be features of the antenna. The antenna in this work was developed with all of these characteristics in mind. It is small in size and has a straightforward two-port shape. The antenna, which was created using readily available commercial materials, is essentially a monopoly antenna. The antenna also provides isolation, a low ECC value, broad performance, and high gain. It is clear that the proposed antenna either offers compact size, broadband or low isolation ECC, or high gain, as compared to the other work listed in Table 1. Due to this discussion and debate, the proposed work is a strong candidate for upcoming 5G applications.

## 4. Conclusions

Two-port MIMO antennas for 28 GHz are presented in this article. Initially, a single element was designed after following three design steps. Afterwards, the two-port MIMO antennas were generated, giving a mutual coupling less than −20 dB. The decoupling patch was loaded between two antenna elements to overcome mutual coupling. After loading the decoupling structure, the antenna offered mutual coupling less than –32 dB. The MIMO antenna presented in this article has a compact size of 25 mm × 10 mm, with a thickness of 1.52 mm. The antenna operates at a wideband of 25.25–29.85 GHz with a peak gain of 10.8 dBi. The antenna offers a good value of ECC, around 0.001, and DG around 9.99 dB. Other MIMO parameters are also analyzed, which are in the acceptable range. The software study of the design and analysis of the antenna was carried out using the EM software tool HFSS (High Frequency Structure Simulator). The hardware prototype was fabricated to validate the simulated outcomes. Moreover, a comparison is provided in the form of table to compare the antenna results with published works. The outcomes and table of comparison make the proposed antenna a potential applicant for future 5G millimeter-wave applications.

## Figures and Tables

**Figure 1 micromachines-14-02065-f001:**
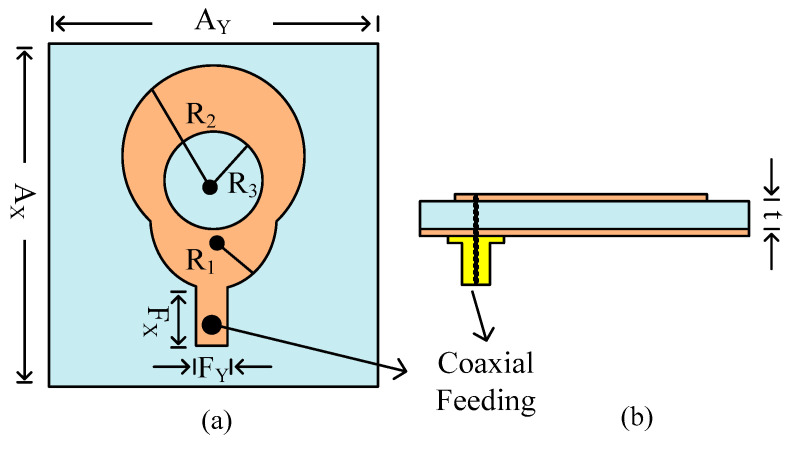
Structural layout of single element of antenna: (**a**) top side and (**b**) side view.

**Figure 2 micromachines-14-02065-f002:**
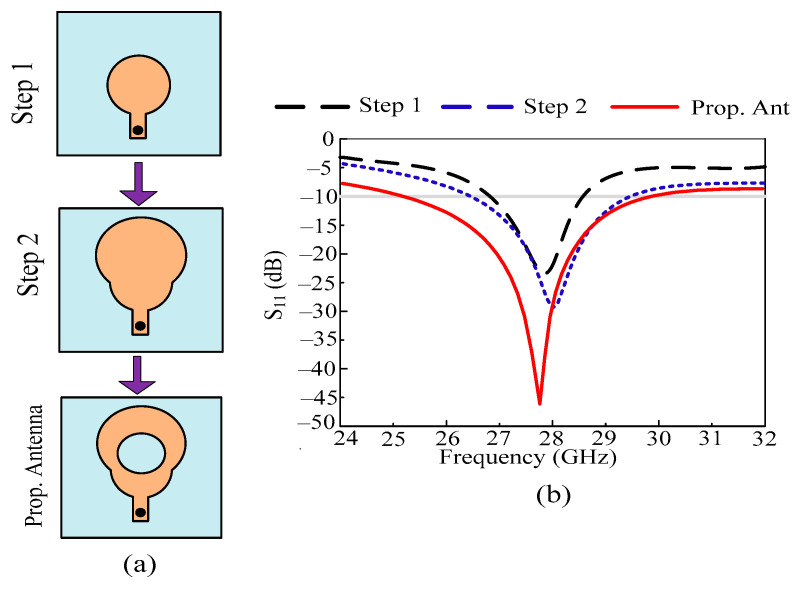
(**a**) Design stages of proposed work; (**b**) S_11_ plot of various stages of antenna design.

**Figure 3 micromachines-14-02065-f003:**
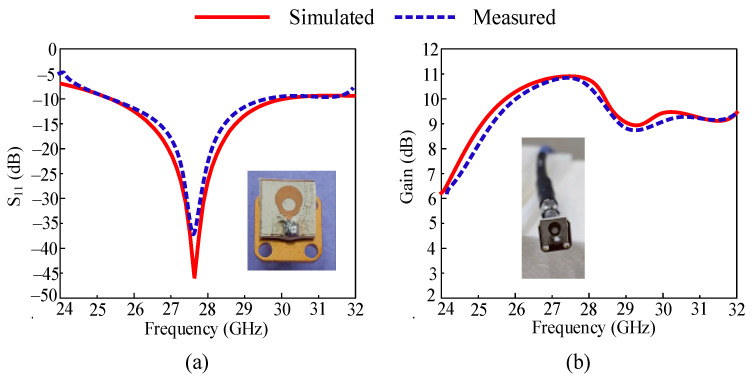
Measured and simulated (**a**) S_11_ curve and (**b**) gain.

**Figure 4 micromachines-14-02065-f004:**
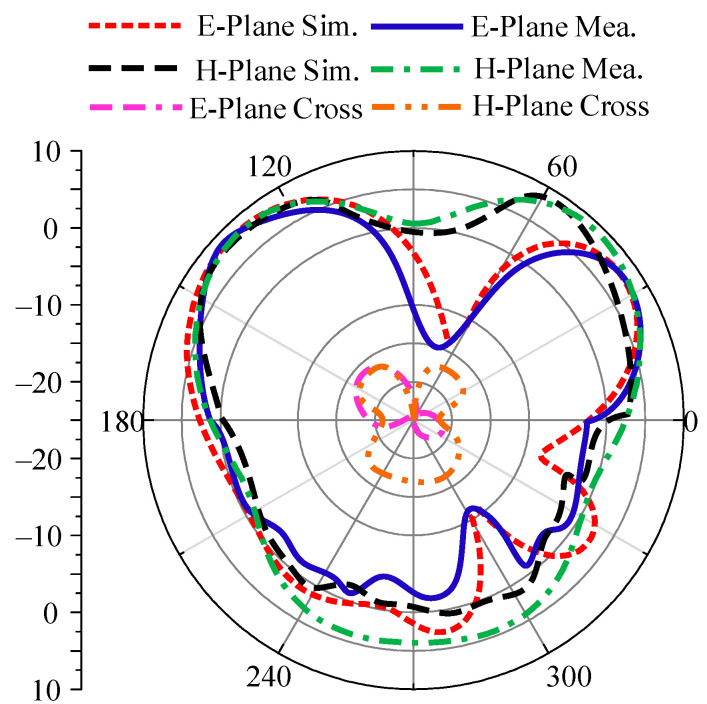
Radiation pattern of proposed antenna at 28 GHz.

**Figure 5 micromachines-14-02065-f005:**
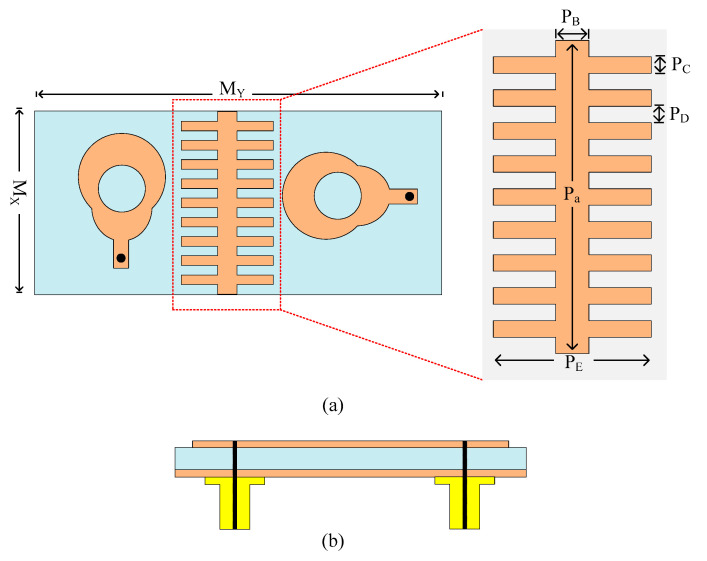
(**a**) Representation of two-port MIMO antenna having a decoupling patch; (**b**) side visual of design to express connectors.

**Figure 6 micromachines-14-02065-f006:**
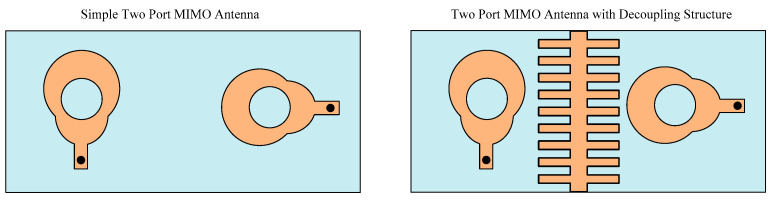
Two port antennae with and without decoupling structure.

**Figure 7 micromachines-14-02065-f007:**
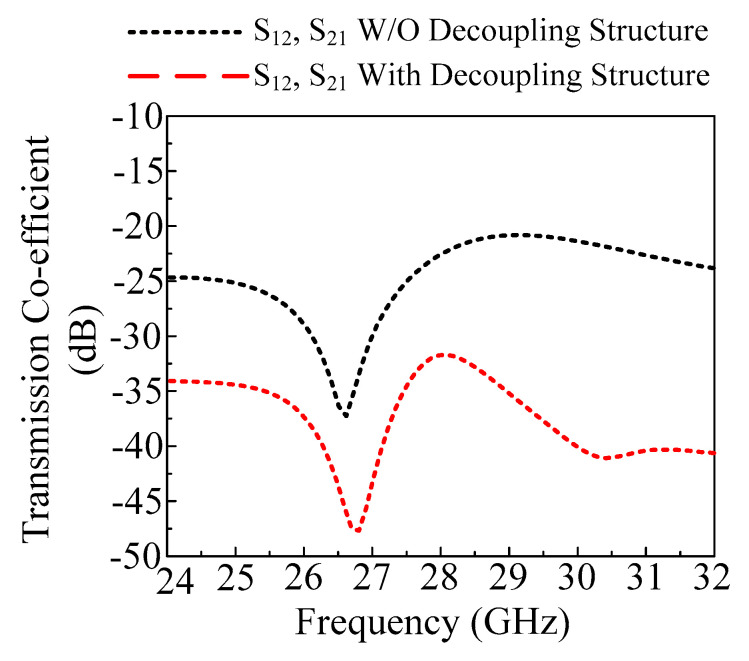
Transmission coefficient of proposed work with and without decoupling structure.

**Figure 8 micromachines-14-02065-f008:**
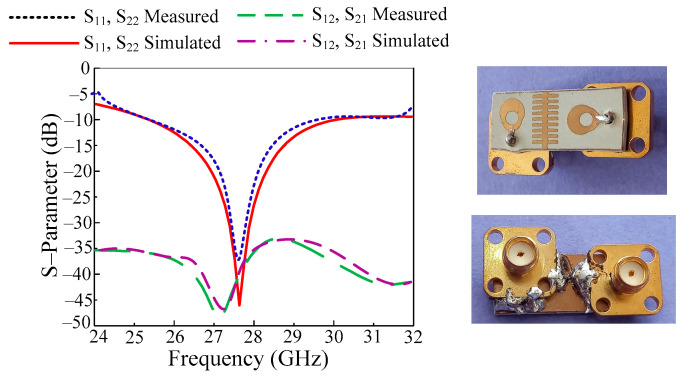
S-parameter of proposed work along with tested results.

**Figure 9 micromachines-14-02065-f009:**
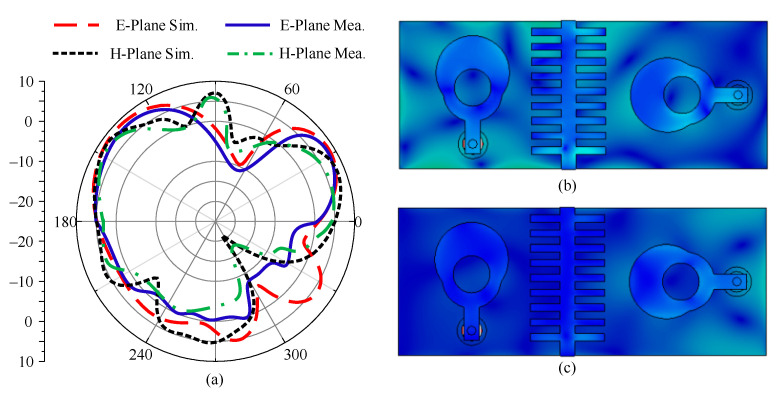
(**a**) Radiation pattern of proposed MIMO antenna at 28 GHz, surface current distribution when (**b**) port 1 is excited, and when (**c**) port-2 is excited.

**Figure 10 micromachines-14-02065-f010:**
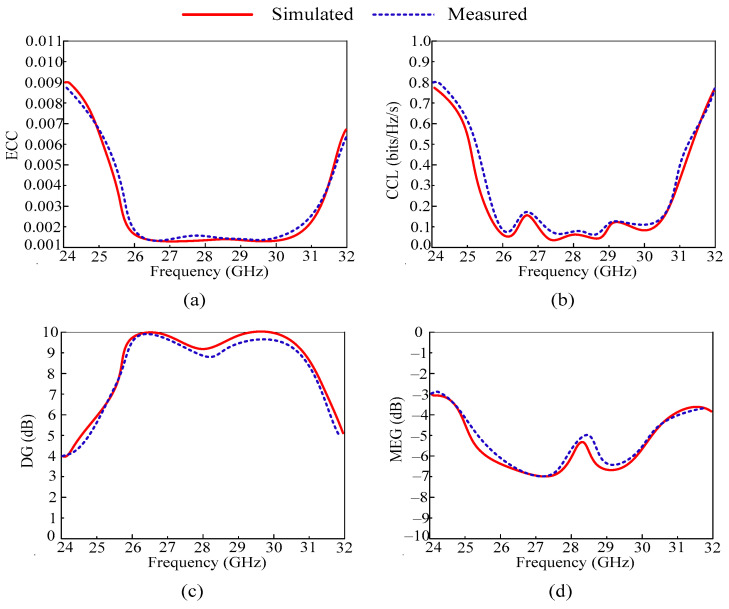
Analysis of important MIMO parameters in terms of (**a**) ECC, (**b**) CCL, (**c**) DG, and (**d**) MEG.

**Table 1 micromachines-14-02065-t001:** Performance comparison of proposed and published work.

Ref	Antenna Size (mm × mm × mm)	Bandwidth (GHz)	Isolation (dB)	Gain (dBi)	ECC	No. of Ports	Antenna Type	Substrate Material	Technique Used
[19]	85 × 21 × 0.508	27–32	38	10.5	–	2	Antipodal Fermi Tapered slot	RO4003	Metamaterial
[20]	26 × 14.5 × 0.508	26.5–29.5	38	4.5	0.0001	2	Monopole Antenna	RT5880	Metamaterial
[28]	20 × 20 × 0.254	27.2–28.5	24	–	0.013	2	DRA	RT5880	Parasitic Patch
[30]	25 × 15 × 0.25	26–29	25	–	0.007	2	DRA	RT5881	Parasitic Patch
[31]	12 × 24 × 1.51	20–28	15	–	–	2	Monopole Antenna	RT5880	Parasitic Patch
This work	25 × 10 × 1.52	25.25–29.85	47	10.8	0.001	2	Monopole Antenna	RT6002	Parasitic Patch

## Data Availability

All the data is available in the study.

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
