# Peer review of "Mutual Coupling Reduction in Compact MIMO Antenna Operating on 28 GHz by Using Novel Decoupling Structure"

_micromachines, 2023, doi:10.3390/mi14112065_

Round 1

Reviewer 1 Report

Comments and Suggestions for Authors

A two-port multiple-input-multiple-output (MIMO) antenna for the mm-wave band is presented in this paper. However, the design proposed by the authors is not new. In the last few years, many authors have presented similar (two-, four-, and eight-port, etc.) MIMO antenna structures in the microwave and mm-wave bands. A previously reported antenna design is replicated in this work by simply modifying the decoupling structure.

The reviewer does not see any significant contribution from the authors' end, and they have not proposed anything useful that can be used practically.

Comments on the Quality of English Language

Minor English language corrections are required.

Reviewer 2 Report

Comments and Suggestions for Authors

This manuscript introduces a MIMO antenna operating at 28 GHz using mutual decoupling techniques. However, certain aspects merit consideration and improvement:

1. Please add all necessary parameters to show the detail dimension in Fig.1 and Fig.5 and the coordinate system.

2. In Fig.2b , Fig. 3a, the absolute values of S11 should be positive.

3 Please add the current distribution figures of two port antenna.

4. Please add the photo of measurement setup such as the anechoic chamber.

5. It seems that the measured lowest value is below -50 dB in Fig.8. Please show this lowest value in this Figure.

6. Why the Simulated E-plane and the measured H-plane patterns only cover 0 to 180 degrees in Fig. 9? Please add the co-pol and the cross-polar radiation patterns.

7. The authors should include all equations to calculate ECC, MEG, DG, and CCL.

8. Please only compare the performance of 2 ports MIMO antennas with this work in Table 1.

Comments on the Quality of English Language

 The manuscript contains grammatical and typographical errors that necessitate correction. Thorough proofreading and editing efforts are recommended to rectify these issues.

Reviewer 3 Report

Comments and Suggestions for Authors

In this manuscript, the authors introduce a compact and simplified antenna design operating at 28 GHz, suitable for integration with 5G and future communication systems. Their work presents an innovative and highly significant contribution to antenna design, particularly in the context of 5G and millimeter-wave communication systems. The novel decoupling structure and outstanding performance metrics make your design a promising candidate for future 5G applications, addressing the need for compact, high-performance antennas. Their results are interesting and solid. The manuscript can be considered for publishing in Micromachines with revisions:

1. The abbreviations used in the Abstract are also need to be marked with corresponding full spells.

2. The authors mention "some imperfections in results." It would be better to further elaborate on these imperfections and their potential impact on the antenna's performance or measurements to improve the transparency to the research.

3. Could the authors provide more detailed figure legends? That will facilitate readers’ direct understanding.

Round 2

Reviewer 1 Report

Comments and Suggestions for Authors

The authors have improved the quality of the manuscript and tried to address the comments of the reviewer.

Comments on the Quality of English Language

Minor English corrections are required.